# Derangements and Reversibility of Energy Metabolism in Failing Hearts Resulting from Volume Overload: Transcriptomics and Metabolomics Analyses

**DOI:** 10.3390/ijms23126809

**Published:** 2022-06-18

**Authors:** Ying-Chang Tung, Mei-Ling Cheng, Lung-Sheng Wu, Hsiang-Yu Tang, Cheng-Yu Huang, Gwo-Jyh Chang, Chi-Jen Chang

**Affiliations:** 1Department of Cardiology, Linkou Chang Gung Memorial Hospital, Taoyuan 33323, Taiwan; n12374@cgmh.org.tw (Y.-C.T.); r5278@cgmh.org.tw (L.-S.W.); 2School of Medicine, College of Medicine, Chang Gung University, Taoyuan 33302, Taiwan; 3Metabolomics Core Laboratory, Healthy Aging Research Center, Chang Gung University, Taoyuan 33323, Taiwan; chengm@mail.cgu.edu.tw (M.-L.C.); tangshyu@gmail.com (H.-Y.T.); chenyu7015@gmail.com (C.-Y.H.); 4Clinical Metabolomics Core Laboratory, Chang Gung Memorial Hospital, Taoyuan 33323, Taiwan; 5Department of Biomedical Sciences, College of Medicine, Chang Gung University, Taoyuan 33323, Taiwan; 6Graduate Institute of Clinical Medicinal Sciences, College of Medicine, Chang Gung University, Taoyuan 33323, Taiwan; gjchang@mail.cgu.edu.tw

**Keywords:** volume overload, compensated hypertrophy, heart failure, energy metabolism

## Abstract

Derangements in cardiac energy metabolism have been shown to contribute to the development of heart failure (HF). This study combined transcriptomics and metabolomics analyses to characterize the changes and reversibility of cardiac energetics in a rat model of cardiac volume overload (VO) with the creation and subsequent closure of aortocaval fistula. Male Sprague–Dawley rats subjected to an aortocaval fistula surgery for 8 and 16 weeks exhibited characteristics of compensated hypertrophy (CH) and HF, respectively, in echocardiographic and hemodynamic studies. Glycolysis was downregulated and directed to the hexosamine biosynthetic pathway (HBP) and *O*-linked-N-acetylglucosaminylation in the CH phase and was further suppressed during progression to HF. Derangements in fatty acid oxidation were not prominent until the development of HF, as indicated by the accumulation of acylcarnitines. The gene expression and intermediates of the tricarboxylic acid cycle were not significantly altered in this model. Correction of VO largely reversed the differential expression of genes involved in glycolysis, HBP, and fatty acid oxidation in CH but not in HF. Delayed correction of VO in HF resulted in incomplete recovery of defective glycolysis and fatty acid oxidation. These findings may provide insight into the development of innovative strategies to prevent or reverse metabolic derangements in VO-induced HF.

## 1. Introduction

In valvular insufficiency, sustained blood flow regurgitation leads to volume overload (VO) and ventricular remodeling, resulting in compensated hypertrophy (CH) in the early phase and progression to heart failure (HF) if left untreated. Chronic severe mitral regurgitation causes an increase in left ventricular (LV) wall stress due to excessive diastolic inflow volume [1]. The LV adapts during a typically long compensated state, marked by eccentric ventricular hypertrophy and enhanced ventricular compliance. After a variable period, the LV progresses from this transitional stage to a chronically decompensated stage [2]. Timely surgical repair or replacement of a dysfunctional valve may lead to favorable reverse remodeling and better clinical outcomes [2,3]. However, if surgery is delayed until significant symptoms develop, LV remodeling may not be reversible and will adversely affect clinical outcomes.

Growing evidence has demonstrated the role of cardiac energy derangements in the pathogenesis of HF [4]. Changes in energy metabolism vary with the etiology and severity of HF, as well as co-existing comorbidities such as diabetes and obesity. It is generally accepted that the major source of substrate utilization shifts from fatty acid oxidation to glucose metabolism as an oxygen-sparing mechanism in the process of cardiac remodeling [5,6,7]. This substrate shift, along with defects in the tricarboxylic acid (TCA) cycle, oxidative phosphorylation, and high-energy phosphate metabolism, eventually results in adenosine triphosphate deficiency and impaired contractility in failing hearts [8,9]. Pharmacological intervention to ameliorate energy and metabolic deficits has become a novel approach for improving cardiac efficiency.

Energy metabolic disturbances of VOed hearts in the transition toward HF are much less studied compared to pressure overload, myocardial ischemia, or dilated cardiomyopathy. Moreover, the reversibility of metabolic remodeling after early versus delayed surgical correction of hemodynamic overload remains unclear. Characterization of cardiac energy derangements in response to VO and metabolic reversibility after corrective surgery may help identify new therapeutic targets for VO-induced HF. The aortocaval fistula (ACF) operation is a commonly used experimental VO model [10,11,12,13]. This study aimed to investigate the changes in and reversibility of cardiac energy metabolism in VOed rat hearts by using a combination of transcriptomics and metabolomics analyses.

## 2. Results

### 2.1. Cardiac Remodeling after the Creation and Subsequent Correction of VO

The echocardiographic and hemodynamic findings are shown in Figure 1A,B, respectively, and summarized in Appendix A. Adult, male, 8–10-week-old Sprague–Dawley (SD) rats subjected to ACF exhibited eccentric hypertrophy, with remarkable chamber dilation relative to the increase in ventricular wall thickness. LV systolic function, as measured by fractional shortening (FS), was preserved at 8 weeks, but reduced significantly at 16 weeks of VO, compatible with the phenotypes of CH and HF. Two other groups of rats with ACF creation underwent echocardiographic and hemodynamic studies at 8 and 16 weeks before being subjected to correction of volume overload by ACF ligation (CH_COV_ and HF_COV_ groups, respectively). Four weeks after ACF closure, the LV and left atrial (LA) diameters in CH and HF rats were slightly reduced and were still significantly larger than those in the corresponding sham controls. Notably, LV fraction shortening in HF rats did not improve after VO correction (47.72%, 35.52%, and 36.21% for controls, HF, and HF_COV_, respectively; *p* < 0.001 for HF_COV_ vs. controls; *p* = 0.772 for HF_COV_ vs. HF). In addition, the elevated LV end-diastolic pressure (LVEDP) and reduced maximal slope of LV systolic pressure increment (+dP/dt_max_) and diastolic pressure decrement (−dP/dt_max_) in failing hearts did not recover at 4 weeks after the correction, although it remains to be determined whether a longer period of observation would result in further improvement. Histological examination of the LV free wall showed that the size of cardiomyocytes determined by cross-sectional area was significantly increased in the CH and HF phases compared with that of the corresponding sham controls, and remained significantly larger after VO correction. Enlargement of the endomyosial and perimyosial space was found in the HF phase. Correction of VO resulted in narrowing of the enlarged endomyosial space but not the perimyosial space (Figure 1C). These findings suggest that the creation and closure of the ACF represent a suitable model for investigating VO-induced cardiac remodeling and reverse remodeling.

### 2.2. Transcriptomics Profiling of Cardiac Energy Metabolism in VO-Induced CH and HF

The histograms of top 30 canonical pathways, ranked by −log (*p* value), in the CH, CH_COV_, HF, and HF_COV_ phases are illustrated in Appendix A, respectively. The Venn diagrams showed that the number of differentially expressed genes involved in glucose and lipid metabolism increased as VOed hearts transitioned from CH to HF (Figure 2C). We validated the results of the microarray analysis by qPCR. Genes related to glycolysis and fatty acid oxidation were selected for the qPCR analysis. The genes that were downregulated in the microarray analysis in at least one phase were validated in all time phases. Microarray and qPCR analyses produced parallel results with respect to the upregulation or downregulation of particular genes, but with some differences in the magnitude of specific changes (Appendix A).

Differentially expressed genes involved in glycolysis, hexosamine biosynthetic pathway (HBP), and fatty acid oxidation are illustrated in Appendix A (CH and CH_COV_) and Figure 3 (HF and HF_COV_) and summarized in Appendix A. *Slc2a4*, the gene responsible for insulin-dependent transportation of glucose to the striated muscle and fat cells (GLUT4), was significantly downregulated in HF rats but not in CH rats. Genes involved in glycolysis, including phosphofructokinase, muscle type, or PFK1 (*Pfkm*), fructose-bisphosphate aldolase B (*Aldob*), muscle-specific phosphoglycerate mutase 2 (*Pgam2*) and enolase 3, beta, muscle (*Eno3*) were downregulated in the HF phase (Figure 3A). 6-Phosphofructo-2-kinase/fructose-2,6-biphosphatase 1 (*Pfkfb1*), the gene encoding 6-phosphofructo-2-kinase/fructose-2,6-biphosphatase 1 (PFK2), which indirectly regulates glycolysis by activating PFK1 through fructose 2,6-bisphosphate, was downregulated in both the CH (Appendix A) and HF phases. The HBP, an accessory pathway of glycolysis, was upregulated in both the CH (Appendix A) and HF phases (Figure 3B), as indicated by the increased expression of glutamine-fructose-6-phosphate transaminase 2 (*GFPT2*), the first and rate-limiting enzyme of the HBP, and UDP-*N*-acetylglucosamine pyrophosphorylase 1 (*UAP1*), the enzyme that generates the end product of HBP (UDP-*N*-acetylglucosamine [UDP-GlcNAc]).

Regarding the genes responsible for fatty acid metabolism, only enoyl-CoA hydratase and 3-hydroxyacyl CoA dehydrogenase (*Ehhadh*), the enzymes catalyzing the second step of mitochondrial β-oxidation, were found to be downregulated in the CH phase (Appendix A). In the HF phase, regulators of lipid metabolism including retinoid X receptor α (*Rxra*) and peroxisome proliferator–activated receptor gamma coactivator-1 beta (*Ppargc1b*), were downregulated. A number of genes encoding enzymes involved in fatty acid metabolism were also downregulated in HF, including the genes of several acyl-CoA synthetases, carnitine-acylcarnitine translocase (*Slc25a20*), and the genes encoding the four enzymes catalyzing mitochondrial β-oxidation (Figure 3C). In contrast, the microarray analysis revealed no significant changes in the TCA cycle (data not shown).

### 2.3. Transcriptomics Profiling of Cardiac Energy Metabolism after Correction of VO

Correction of VO in the CH phase largely reversed the differential expression of the genes involved in glycolysis and HBP (Appendix A). After correction of VO in HF, *Pfkfb1, Aldob,* and *Acaa2* remained significantly downregulated (Figure 3A,C). A trend toward the downregulation of *Ehhadh* was observed in HF (fold change 0.69, *p* = 0.015; Appendix A). Both *Pfkfb1* and *Aldob* regulate glycolysis, whereas *Ehhadh* and *Acaa2* encode enzymes that catalyze two of the four steps of mitochondrial fatty acid oxidation. These findings indicate that derangements of glycolysis and fatty acid oxidation were not completely reversed if VO correction was delayed after HF development.

### 2.4. Metabolomics Profiling of Cardiac Energy Metabolism in VO-induced CH and HF

Z-score plots illustrate the changes in the myocardial levels of metabolites in the different VO phases. The levels of 2-phosphoglycerate, 3-phosphoglycerate, and pyruvate were significantly reduced in both the CH and HF phases and phosphoenolpyruvate were significantly reduced in the HF phase compared to that of the corresponding sham controls (Figure 4A; Appendix A). Notably, compared to that of the CH phase, a trend toward lower levels of glycolytic intermediates in the HF phase was found, with a significant reduction in the levels of fructose 6-phosphate, 2-phosphoglycerate, and 3-phosphoglycerate, suggesting that glycolysis was further suppressed in the HF phase. In contrast, the levels of UDP-GlcNAc, the end product of HBP, significantly increased during the transition from CH to HF, accompanied by an even lower level of fructose 6-phosphate (Figure 4B; Appendix A). These findings indicate an increased HBP flux during progression to HF.

Alterations in fatty acid oxidation in VOed hearts were inferred from the assessment of myocardial carnitine and acylcarnitine levels. In the CH phase, the levels of carnitine and acylcarnitines did not change significantly compared to those in the sham controls (Figure 4C; Appendix A). In contrast, carnitine was significantly reduced and various species of acylcarnitines were significantly increased in the HF phase, suggesting dysregulated fatty acid metabolism. In contrast, the levels of TCA cycle intermediates did not change significantly in either the CH or HF phases compared to that of corresponding sham controls (Figure 4D; Appendix A).

### 2.5. Metabolomics Profiling of Cardiac Energy Metabolism after Correction of VO

In general, the reduced levels of glycolysis intermediates in the VOed hearts increased at 4 weeks after correction of VO, except that the levels of phosphoenolpyruvate in the CH phase and 2-phosphoglycerate and 3-phosphoglycerate in the HF phase remained significantly lower than those in the sham controls (Figure 4A; Appendix A). Although delayed correction of VO in the HF phase significantly increased the level of pyruvate, it remained lower than that in the sham controls. In contrast, the levels of UDP-GlcNAc (Figure 4B; Appendix A) and several acylcarnitines (Figure 4C; Appendix A) decreased after the correction of VO, but remained significantly higher than those in the sham controls, suggesting a sustained increase in HBP flux and defective fatty acid oxidation after correction of VO in the HF phase. Changes in the intermediates of the TCA cycle were unremarkable after VO correction (Figure 4D; Appendix A).

### 2.6. Elevation of Protein O-linked-N-acetylglucosaminylation (O-GlcNAcylation)

Since we found an increased flux to the HBP in both the CH and HF phases and post-correction of VO in the HF phase, we assayed the myocardial pattern of protein *O*-GlcNAcylation in each phase by immunoblotting by using a pan-specific CTD 110.6 anti-*O*-GlcNAc antibody. The patterns varied among different phases. A number of bands showed increased *O*-GlcNAcylation labeling in different VO phases compared to the corresponding sham controls (Figure 5A). A significant increase in the level of global protein *O*-GlcNAcylation was observed in VOed rats in both the CH and HF phases and VOed rats with VO correction in the HF phase (Figure 5B).

## 3. Discussion

This study combined transcriptomics and metabolomics analyses to investigate the dynamic changes in cardiac energy metabolism during the development of HF induced by VO and the reversibility of the changes after correction of VO. The major findings were as follows: (1) glycolysis was downregulated in the CH phase and was further suppressed in HF with increased flux to the HBP in both phases; (2) dysregulation of fatty acid oxidation was not found in the CH phase but was prominent after progressing to the HF phase; (3) early correction of VO in the CH phase largely reversed the differential expression of genes involved in glycolysis, HBP, and fatty acid oxidation with normalization of the levels of most metabolic intermediates; and (4) delayed correction of VO in the HF phase showed incomplete reversion of the differential expression of genes involved in glycolysis and fatty acid oxidation as well as incomplete recovery of reduced glycolytic intermediates and increased end product of the HBP and acylcarnitines, suggesting persistent defective glycolysis and fatty acid oxidation after delayed correction of VO.

### 3.1. Downregulation of Glycolysis in Chronic Volume-Overloaded Hearts in CH and HF Phases

Most studies on animal models of HF and patients with HF have found that the source of energy shifts from fatty acids to glucose during HF development. Because glucose has a higher oxygen efficiency for ATP production, this shift in energy source was thought to be an oxygen-sparing mechanism in the process of cardiac remodeling [8,14]. Studies on transverse aortic constriction-induced pressure overload demonstrated that myocardial glucose uptake and utilization rates were increased and fatty acid oxidation was suppressed. As observed in the fetal hearts, increased glucose utilization in pressure overload was associated with an increase in insulin-independent GLUT1 and decreased insulin-dependent GLUT4 transporters [15,16]. However, in this study, we found that upon exposure to chronic VO, the heart exhibited suppressed glucose uptake and glycolysis, particularly in the HF phase, indicating that the pattern of energy metabolic derangements caused by VO is distinct from those caused by other etiologies. In contrast to an upregulation of GLUT1 in pressure-overloaded models, GLUT1 gene expression did not changed significantly in this study. The differences in experimental models and hemodynamic loads may account for the different metabolic adaptations between the pressure-overloaded and VOed hearts. *Slc2a4*, the gene encoding the insulin-regulated glucose transporter GLUT4, was downregulated during the HF phase. Mice with deletion of cardiac GLUT4 have been shown to exhibit CH at baseline and severe contractile dysfunction in response to pressure overload, suggesting an association between the loss of insulin-dependent glucose uptake and pathological remodeling [17,18]. Genes encoding PFK-1, the rate-limiting enzyme of glycolysis, and PFK-2, the enzyme responsible for the synthesis of fructose 2,6-bisphosphate and indirect regulation of glycolysis, were also downregulated in the HF phase. Notably, transcription of PFK-2 was suppressed in the CH phase and was not reversed by VO correction after progression to HF. Mice with either kinase- or phosphatase-deficient PFK-2 have been demonstrated to develop cardiac hypertrophy at baseline [19,20], suggesting that defective PFK-2 activity per se can drive hypertrophy. Phosphorylation and activation of PFK-2 stimulate glycolysis and improves cardiac energetics [19,21,22] and, therefore, may serve as a novel therapeutic approach for the metabolic dysfunction in HF resulting from VO.

### 3.2. Upregulation of the HBP and Protein O-GlcNAcylation in VO

HBP and subsequent protein *O*-GlcNAcylation were shown to be activated in response to cellular stresses or during the development of cardiac hypertrophy [23,24]. We showed, in the VOed heart, the increased transcription of GFAT2 and elevated cardiac UDP-GlcNAc levels in both the CH and HF phases. The role of *O*-GlcNAcylation is multifaceted and varies depending on the stimulus type. An acute increase in *O*-GlcNAcylation has been considered a recruitable mechanism of cardioprotection, particularly in ischemia-reperfusion injury [25,26]. However, excessive activation has been shown to contribute to contractile and mitochondrial dysfunction, and attenuation of *O*-GlcNAcylation has been demonstrated to reverse cardiac remodeling and improve myocardial contractility in diabetic cardiomyopathy [27,28] and pressure-overloaded hearts [29]. The persistent increase in UDP-GlcNAc levels and an increase in protein *O*-GlcNAcylation after delayed correction of VO in the HF phase suggest that *O*-GlcNAcylation may play a role in the irreversibility of cardiac dysfunction and adverse remodeling through global posttranslational modification. 

### 3.3. Dysregulation of Fatty Acid Metabolism in VO-Induced HF

In the VOed heart, we found dysregulation of fatty acid metabolism in the HF phase. Downregulation of genes encoding RXRα and PGC-1β in HF rats suggests defective fatty acid metabolism at the regulator level. Reduced expression of genes encoding acyl-CoA synthetases and carnitine-acylcarnitine translocase also results in insufficient fatty acid metabolism [30]. The decreased transcription of carnitine-acylcarnitine translocase result in the sequestration of carnitine in the form of acylcarnitines, accounting for the reduced carnitine and elevated acylcarnitines in the HF phase. Importantly, genes encoding enzymes catalyzing mitochondrial fatty acid oxidation were also downregulated, which might further increase the levels of acylcarnitines. It has been proposed that the accumulation of incomplete fatty acid oxidation products, such as acylcarnitines, may activate proinflammatory pathways [31] and contribute to insulin resistance [32,33,34]. Furthermore, both myocardial and circulating levels of acylcarnitines were elevated in patients with HF [35,36,37], and circulating acylcarnitines have been shown to predict adverse clinical outcomes [38].

We showed that acetylcarnitine (C2), which is in equilibrium with and therefore a surrogate measurement of acetyl-CoA [39], was elevated in the HF phase, consistent with the finding in patients with HF [40,41] and in a mouse model of pressure overload [42]. Oxidation of lipids and glucose by cardiomyocytes funnels into acetylcarnitine [43,44]; hence, its accumulation indicates a “metabolic bottleneck” that generation of acetyl-CoA from myocardial substrate utilization may have exceeded the capacity for entry into the TCA cycle. In addition, acetylcarnitine has been recognized as a metabolic indicator of ketone utilization in human and animal models of HF [41,45].

### 3.4. No Significant Change in the Gene Expression and Intermediates of the TCA Cycle

Reduced TCA cycle intermediates have been shown in the failing hearts of human and animal models, which may precede the deterioration of systolic function [42,46,47,48]. However, proteomic studies have reported inconsistent results, with reduced protein levels of enzymes of the TCA cycle in some studies [49,50,51] and elevated or unchanged in others [52,53]. Both the transcription of enzymes involved in the TCA cycle and the intermediates were not significantly altered in this study. Given impaired glycolysis and FAO, TCA intermediates may derive from anaplerosis or the breakdown of ketone bodies or amino acids. Because the steady-state concentrations of TCA intermediates are typically low compared to the fluxes through the TCA cycle [54], further isotope tracer studies are needed to determine the activity of TCA cycle in the VOed heart.

### 3.5. Reversibility of Dysregulated Energy Metabolism after the Correction of VO

The reversibility of energy derangement of the failing heart resulting from hemodynamic stress after correction of hemodynamic stress remains unclear. In this study, we investigated the effects of early and delayed VO correction on energy metabolic reversibility. Transcriptomics analysis showed that early correction of VO in the CH phase reversed most differentially expressed genes involved in glycolysis and HBP. In contrast, after delayed correction in the HF phase, *Pfkfb1, Aldob,* and *Acaa2* remained significantly downregulated and there was a strong trend toward the downregulation of *Ehhadh*. Both *Pfkfb1* and *Aldob* regulate glycolysis and *Ehhadh* and *Acaa2* encode enzymes that catalyze two of the four steps of mitochondrial fatty acid oxidation. Consistent with this expression profile, metabolomics studies showed decreased levels of 2-phosphoglycerate, 3-phosphoglycerate and pyruvate, and increased levels of a number of acylcarnitine specimens. These findings indicate that delayed correction of VO after the heart has progressed to the HF phase does not completely reverse the derangements of glycolysis and fatty acid oxidation, which may contribute to the irreversibility of adverse remodeling and cardiac dysfunction.

### 3.6. Study Limitations

This study had several limitations. Although we combined transcriptomics and metabolomics analyses to investigate VO-induced energy metabolic derangements, our results may not provide adequate information on pathway fluxes, which are typically assessed by using radioactive tracer metabolites (i.e., metabolic flux analysis). We could not specifically determine the changes in glycolysis and glucose oxidation rates in failing hearts. Whether there is uncoupling of glycolysis and glucose oxidation, which has been considered to contribute to HF progression, is unknown. Due to technical limitations, several important metabolites, such as pyridine nucleotides, adenosine triphosphate, and malonyl-coenzyme A, were not assessed in this study. Other important metabolic pathways in HF, such as ketone body oxidation and branched-chain amino acid metabolism, have not yet been investigated. Although HBP was activated in this study, how subsequent *O*-GlcNAcylation may have contributed to structural and metabolic remodeling remains to be explored. Myocardial tissue from the LV was used for all experiments, and we were unable to distinguish between the metabolic perturbations originating from cardiac myocytes and those from non-myocytes. Although the present study provided static snapshots of the cardiac metabolic states in different phases of VO, the study design could not determine whether the derangements in cardiac energetics play a causative role in the pathogenesis of HF, or are merely a responsive epiphenomenon that accompanies structural remodeling. We did not assess the changes in the feeding behavior of VOed rats. How the changes in feeding behavior during HF development may have influenced the study results is unknown.

## 4. Methods and Materials

### 4.1. Animal Surgeries for the Creation and Correction of Cardiac VO

An ACF model was created in adult, male, 8–10-week-old SD rats weighing 300–350 mg [55]. After anesthetization with ketamine (50 mg/kg) and xylazine (8 mg/kg), midline abdominal laparotomy was performed to expose the abdominal aorta and inferior vena cava (IVC), and a clamp was placed across both vessels. The abdominal aorta was punctured with an 18-gauge needle. The needle was inserted into the lateral wall of the aorta and advanced medially through the medial wall and into the IVC. The needle was then gently removed and the puncture site was sealed with cyanoacrylate glue. Creation of the ACF was confirmed by visualizing a mixture of bright red arterial blood in the IVC. In the sham controls, the same procedure was performed, except that no punctures were created. Prior studies have shown that rats subjected to ACF develop CH at around 8 weeks and decompensated HF at 16–21 weeks [10,11,12,13]. This study also aimed to investigate the metabolic adaptation to reverse cardiac remodeling after VO correction. Two other groups of SD rats with ACF creation underwent echocardiographic and hemodynamic studies at 8 and 16 weeks before being subjected to correction of VO by ACF ligation (CH_COV_ and HF_COV_ groups, respectively). ACF ligation was performed under anesthesia as described previously. After midline abdominal laparotomy, the ACF was exposed and ligated by using a hemoclip, and the abdominal wall was closed. The surgical procedures of the creation and closure of an ACF are illustrated in Figure 6. When the rats were sacrificed, ketamine (50 mg/kg) and xylazine (8 mg/kg) were administered intraperitoneally for anesthesia. All animal experimental protocols were reviewed and approved by Chang Gung University IACUC (CGU107-192).

### 4.2. EchocarDiography and Invasive Hemodynamic Assessment

Echocardiography and hemodynamic assessment were performed at 8 and 16 weeks post-ACF. Briefly, the animals were anesthetized with halothane inhalational anesthesia, and echocardiography was performed with multiple views by using a 10 MHz ultrasound probe (Vivid System 5, GE, Boston, MA, USA). The following parameters were collected: heart rate, LV end-diastolic diameter, end-systolic diameter, LV FS, thickness of the interventricular septum and LV posterior wall, and LA diameter. For invasive hemodynamic study, animals were anesthetized with ketamine (50 mg/kg) and xylazine (8 mg/kg) and a microtip pressure–volume catheter (SPR-838, 2.0 F, Millar Instruments, Houston, TX, USA) was inserted into the LV via the right carotid artery under pressure guidance. Pressure and 3-lead ECG signal were digitalized at 1 kHz and recorded for off-line analysis by using LabChart software (ADInstruments, Dunedin, New Zealand). The following data were recorded and analyzed off-line by using a cardiac pressure–volume analysis program (PVAN 3.6, Millar Instruments, Houston, TX, USA): heart rate, maximal LV systolic pressure, LVEDP, +dP/dt_max_, and −dP/dt_max_.

### 4.3. Transcriptomics Analysis of Energy Metabolic Pathways

After sacrifice, the LV was excised, rinsed in phosphate buffered saline, and freeze-clamped in liquid nitrogen. Total RNA was isolated from powdered LV by using the TRIzol Reagent (Invitrogen, Carlsbad, CA, USA) and further purified and concentrated by using the WelPrep tissue RNA kit (Welgene, Taipei, Taiwan). The quality and quantity of total RNA were analyzed by using a Bioanalyzer 2100 (Agilent Technologies, Santa Clara, CA, USA). Gene expression profiles in tissue RNA were analyzed by using the Affymetrix GeneChip Rat Gene 1.0 ST array (Santa Clara, CA, USA), following the manufacturer’s protocol. Microarray experiments were performed at the Genomics Medicine Research Core Laboratory of Chang Gung Memorial Hospital, Taiwan. The data, including all raw microarray data, were deposited in the Gene Expression Omnibus database with accession no. GSE97044 (https://www.ncbi.nlm.nih.gov/geo/query/acc.cgi?acc=GSE97044 (accessed on 15 May 2022)). Genes with fold change > 1.5 and *p* < 0.05 in VOed rats compared with those in the sham controls were considered differentially expressed. The array data were uploaded into Ingenuity Pathway Analysis (QIAGEN IPA 22.0, Redwood City, CA, USA) for pathway enrichment analyses and to construct Venn diagrams to illustrate the transcriptomics profiles of differentially expressed genes involved in glucose and lipid metabolism in different phases of VO. The differentially expressed genes were further mapped to specific metabolic pathways in IPA, including glycolysis, HBP, mitochondrial fatty acid oxidation, and TCA cycle.

### 4.4. Targeted Metabolomics Analysis

Liquid chromatography/time-of-flight mass spectrometry (LC/TOF-MS) analysis was performed as previously described [56] to identify and determine the myocardial levels of metabolites involved in specific metabolic pathways. Liquid chromatographic separation was achieved on a 100 mm × 2.1 mm Acquity 1.7-μm C18 column (Waters Corp, Milford, MA, USA) by using an ACQUITY Ultra Performance Liquid Chromatography^TM^ system (Waters). The column was maintained at 45 °C and a flow rate of 0.25 mL/min. Samples were eluted from the LC column by using linear gradients of solvent A (2 mM ammonium formate) and solvent B (100% acetonitrile): 0–2.5 min, 1–48% B; 2.5–3 min, 48–98% B; 3–4.2 min, 98% B; and 4.3–6 min, 1% B for re-equilibration. Mass spectrometry was performed on an Agilent Q TOF (6510 Q-TOF MS; Agilent Technologies, Santa Clara, CA, USA) operated in electrospray positive-ion (ESI+) and electrospray negative-ion (ESI−) modes. The scan range was 20 to 990 *m/z*. The capillary and cone voltages were set to 3000 and 35 V, respectively. All analyses were performed by using the lock spray to ensure accuracy and reproducibility; sulfadimethoxine was used as the lock mass at a concentration of 60 ng/mL and a flow rate of 6 L/min (an [M + H]^+^ ion at 311.0814 Da in ESI positive mode and an [M + H]^−^ ion at 309.0658 Da in ESI negative mode). Metabolite concentrations were assessed as previously described [57]. Metabolite concentrations were calculated and expressed as ppb or μM. Values below the limit of detection or below the lower limit of quantification were excluded.

### 4.5. Real-Time Quantitative Reverse Transcription-Polymerase Chain Reaction (qPCR)

Total RNA was extracted by using an RNeasy Fibrous Tissue Mini Kit (QIAGEN, Redwood City, CA, USA). The quality and quantity of the total RNA were analyzed by using a Bioanalyzer 2100 system (Agilent Technologies, Santa Clara, CA, USA). The oligonucleotide sequences of the specific primers used in the PCR reaction are shown in Appendix A.

### 4.6. Western Blot Analysis

Specimens of the LV free wall were homogenized by freezing and grinding. Total proteins were extracted by the RIPA Lysis Buffer, and then the proteins were resolved and separated by 8% sodium dodecyl sulfate-polyacrylamide gel electrophoresis and transferred to polyvinylidene fluoride membranes (Merck Millipore, Darmstadt, Germany). The membranes were incubated with a pan-specific monoclonal anti-O-GlcNAc (CTD 110.6) antibody (Cell Signaling Technology, Danvers, MA, USA) as the primary antibodies. Signals were detected by using the enhanced chemiluminescence detection method (Cytiva Amersham ECL, Marlborough, MA, USA) and quantified relative to GAPDH by using densitometry.

### 4.7. Statistical Analysis

Data are presented as mean ± standard deviation. Differences between two groups were determined by using an unpaired *t*-test with GraphPad Prism 9.1.1 (San Diego, CA, USA). For multiple groups, one-way ANOVA with Bonferroni’s post hoc test was applied to compare the data. Statistical significance was set at *p* < 0.05.

## 5. Conclusions

In this VO rat model, combined transcriptomics and metabolomics analyses showed that glycolysis was downregulated and directed toward HBP in the CH phase. During progression to HF, glycolysis was further suppressed and fatty acid oxidation was defective, as indicated by acylcarnitine accumulation. Correction of VO largely reversed the differential expression of genes involved in glycolysis, HBP, and fatty acid oxidation in CH but not in HF. Delayed correction of VO resulted in persistent derangements in glycolysis and fatty acid oxidation. These findings may provide insight into the development of innovative strategies to prevent or to reverse metabolic derangements in VO-induced HF.

## Figures and Tables

**Figure 1 ijms-23-06809-f001:**
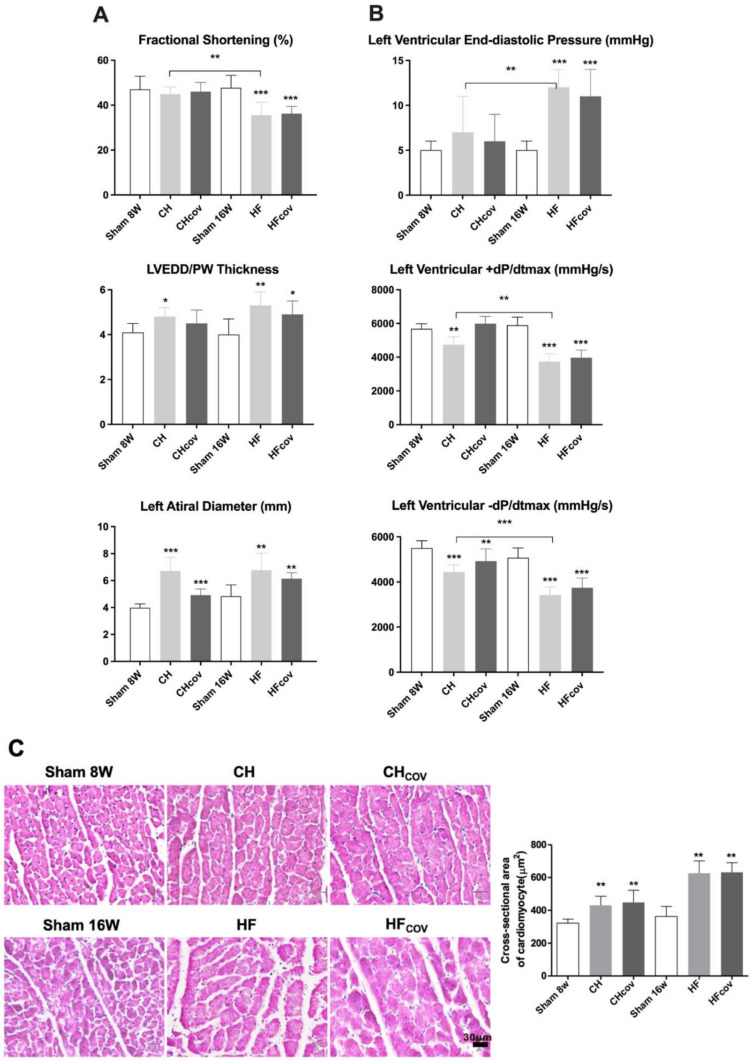
Characterization of rats with volume overload (VO) or with correction of VO. Echocardiography (**A**) and hemodynamic assessment (**B**) of rats with ACF-induced VO in the compensated hypertrophic (CH) and heart failure (HF) phases, VOed rats with correction of VO in the CH (CH_COV_) and HF phases (HF_COV_), and the corresponding sham controls. (**C**) Hematoxylin and eosin-stained sections of the left ventricles in different VO phases. The size of cardiomyocytes quantified by the cross-sectional area was compared between VOed rats and corresponding sham controls. *n* = 6 per group. White, light gray, and dark gray columns represent sham controls, VOed rats, and VOed rats with correction of VO, respectively. *, **, and *** denote *p* < 0.05, *p* < 0.01, and *p* < 0.001, respectively. LVEDD, left ventricular end-diastolic diameter; PW, posterior wall.

**Figure 2 ijms-23-06809-f002:**
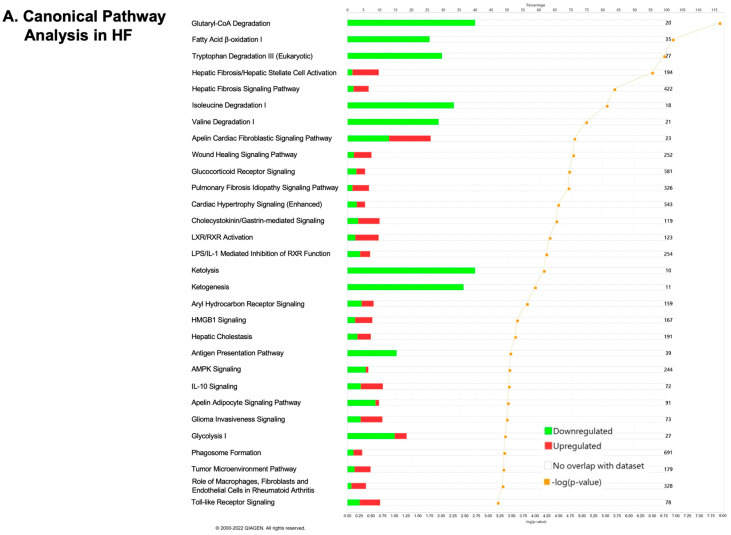
Transcriptomics analyses of rats with volume overload (VO) or with correction of VO. Pathway enrichment analyses of the differentially expressed genes (fold change > 1.5 and *p* < 0.05; ranked by −log [*p* value] of each pathway) in VOed rats in the heart failure (HF) phase (**A**) and in VOed rats with correction of VO in the HF phase (HF_COV_) (**B**); Venn diagrams (**C**) illustrating the number of differentially expressed genes involved in glucose and lipid metabolism in VOed rats in the compensated hypertrophic (CH)and HF phases and VOed rats with correction of VO in the CH (CH_COV_) and HF phases (HF_COV_). *n* = 4 per group.

**Figure 3 ijms-23-06809-f003:**
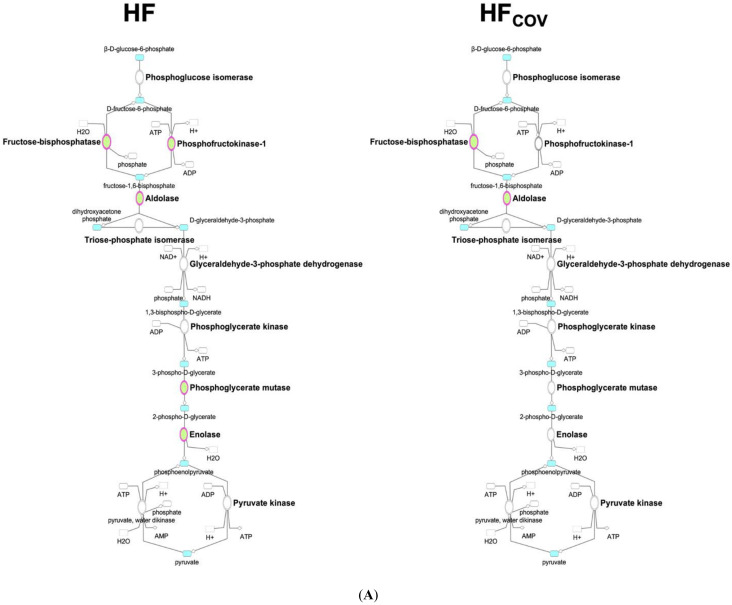
Differential expression of genes (fold change > 1.5 and *p* < 0.05) involved in glycolysis (**A**), hexosamine biosynthetic pathway (**B**), and fatty acid oxidation (**C**) in volume-overloaded rats in the heart failure (HF) phase and in volume-overloaded rats after correction of volume overload in the HF phase (HF_COV_). The ovals in the maps represent enzymes in each metabolic pathway, with red and green denoting upregulation and downregulation of gene expression, respectively. *n* = 4 per group.

**Figure 4 ijms-23-06809-f004:**
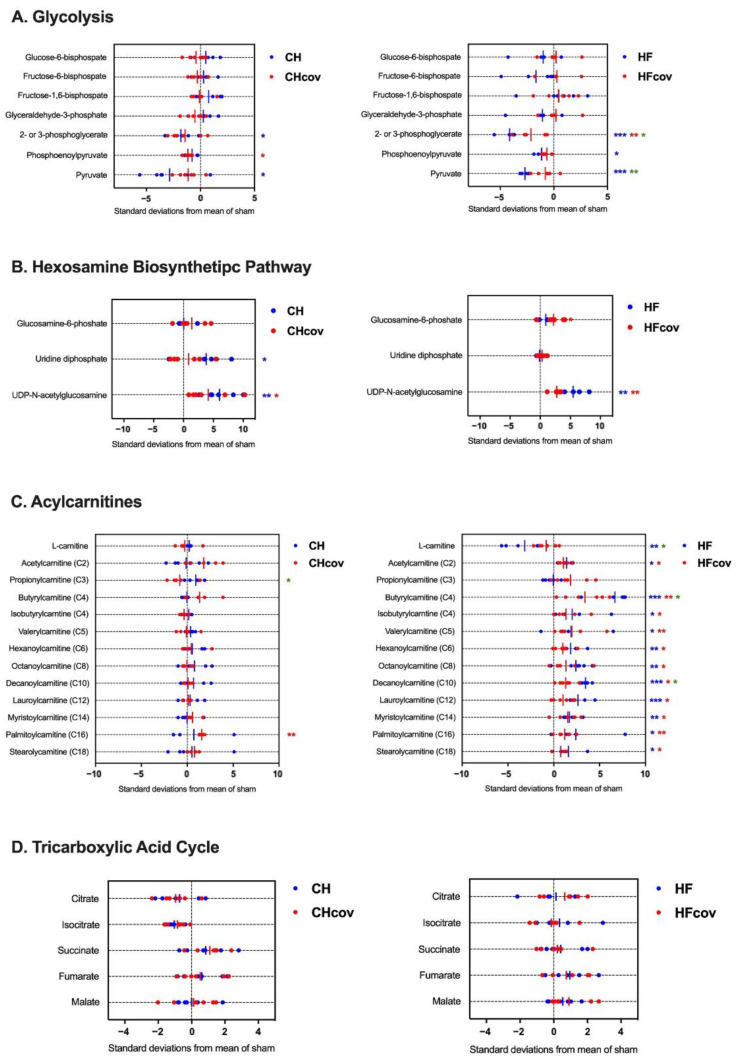
Changes in the myocardial levels of (**A**) glycolysis intermediates, (**B**) hexosamine biosynthetic pathway intermediates, (**C**) acylcarnitines, and (**D**) tricarboxylic acid cycle intermediates in volume-overloaded rats in the compensated hypertrophic (CH) and heart failure (HF) phases and in volume-overloaded rats with correction of volume overload in the CH (CH_COV_) and HF phases (HF_COV_) compared to those in the corresponding sham controls. In each z-score plot, the data are shown as standard deviation from the mean of sham. Each dot represents a single metabolite in each sample. The vertical lines represent the means of z-scores. *n* = 6 per group. Blue, red, and green stars denote comparison between CH or HF vs. sham, CH_COV_ or HF_COV_ vs. sham, and CH vs. HF, respectively. *, **, and *** denote *p* < 0.05, *p* < 0.01, and *p* < 0.001, respectively.

**Figure 5 ijms-23-06809-f005:**
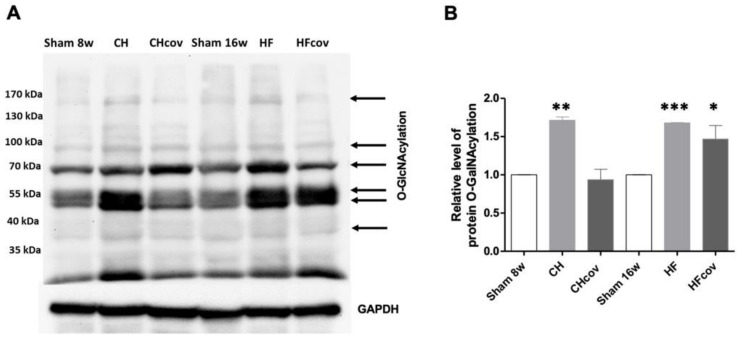
Patterns of protein *O*-GlcNAcylation in the left ventricles of volume-overloaded rats in the compensated hypertrophic (CH) and heart failure (HF) phases and in volume-overloaded rats with correction of volume overload in the CH (CH_COV_) and HF phases (HF_COV_) were analyzed by immunoblotting by using a pan-specific CTD 110.6 anti-*O*-GlcNAc antibody. Arrows indicate bands that show altered *O*-GlcNAcylation labeling in volume-overloaded rats compared with the corresponding sham controls (**A**). The levels of protein *O*-GlcNAcylation were quantified relative to glyceraldehyde-3-phosphate dehydrogenase (GAPDH) by densitometry (**B**). *n* = 4 per group. *, **, and *** denote *p* < 0.05, *p* < 0.01, and *p* < 0.001, respectively.

**Figure 6 ijms-23-06809-f006:**
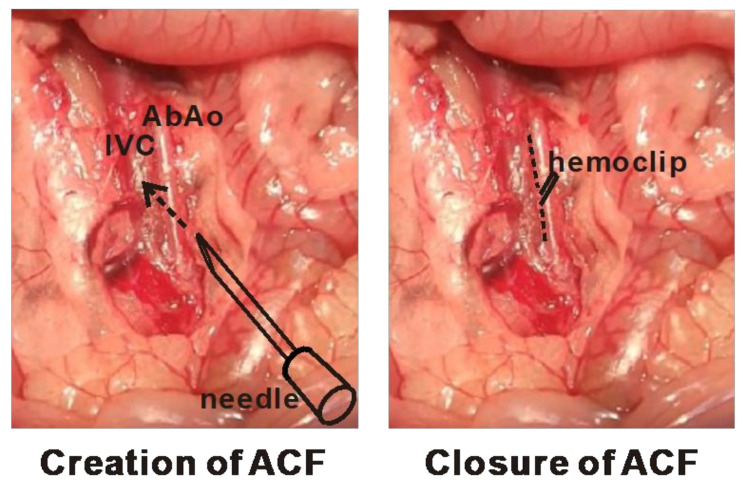
Schematic diagrams showing the procedures to create and to close an aortocaval fistula (ACF). The abdominal aorta (AbAo) was punctured with an 18-gauge needle. The needle was inserted into the lateral wall of the aorta and advanced medially through the medial wall and into the inferior vena cava (IVC) (**left panel**). To close an ACF, the fistula was identified after dissecting the soft tissue between the abdominal aorta and the IVC (dash line) and then clamped with a hemoclip (**right panel**).

## Data Availability

The data presented in this study are available on request from the corresponding author.

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
