# Peer review of "Derangements and Reversibility of Energy Metabolism in Failing Hearts Resulting from Volume Overload: Transcriptomics and Metabolomics Analyses"

_ijms, 2022, doi:10.3390/ijms23126809_

Round 1
Reviewer 1 Report
The article “Derangements and Reversibility of Energy Metabolism in the 2 Failing Hearts Resulting from Volume Overload: Tran-3 scriptomic and Metabolomic Analyses” of Ying-Chang Tung et al. describes the changes and reversibility of cardiac energy metabolism in VOed rat hearts with the combination of transcriptomic and metabolomic analyses. The paper is well presented and well written. It is important to note that the presented results can be used in the development of innovative strategies to prevent or reverse metabolic disorders in heart failure. However, there are minor comments on submitting the article.
1) Line 68… You must specify the age of the rats
2) Figure 5. Y-axis is not labeled.
Too little description of the Figure. No control of protein load.
It is necessary to describe in detail how the samples for electrophoresis were prepared. What percentage of shell was used in electrophoresis? What membrane was used to transfer proteins from the gel to the membrane?
It is necessary to briefly describe the changes in the protein used in the study.
Point out why there are so many major bands in the Western blot?
Author Response
Reviewer 1
The article “Derangements and Reversibility of Energy Metabolism in the 2 Failing Hearts Resulting from Volume Overload: Transcriptomic and Metabolomic Analyses” of Ying-Chang Tung et al. describes the changes and reversibility of cardiac energy metabolism in VOed rat hearts with the combination of transcriptomic and metabolomic analyses. The paper is well presented and well written. It is important to note that the presented results can be used in the development of innovative strategies to prevent or reverse metabolic disorders in heart failure. However, there are minor comments on submitting the article.
Point 1. Line 68… You must specify the age of the rats
Ans: We have clarified in the manuscript that the age of the rats was 8–10 weeks (line 68 in the revised manuscript).
2) Figure 5. Y-axis is not labeled.
Too little description of the Figure. No control of protein load.
It is necessary to describe in detail how the samples for electrophoresis were prepared. What percentage of shell was used in electrophoresis? What membrane was used to transfer proteins from the gel to the membrane?
It is necessary to briefly describe the changes in the protein used in the study.
Point out why there are so many major bands in the Western blot?
Ans: We have revised the method, figure, and figure legend of the immunoblotting in the manuscript. In brief, specimens of LV free wall were homogenized by freezing and grinding. Total proteins were extracted by the RIPA Lysis Buffer, and then the proteins were resolved and separated by 8% sodium dodecyl sulfate-polyacrylamide gel electrophoresis and transferred to polyvinylidene fluoride membranes (Merk Millipore, Netherlands). The membranes were incubated with a pan-specific monoclonal anti-O-GlcNAc (CTD 110.6) antibody (Cell Signaling Technology, Danvers, Massachusetts, USA) as the primary antibodies. Signals were detected using the enhanced chemiluminescence detection method (Amersham, Netherlands) and quantified relative to GAPDH using densitometry (line 158–167).
We assayed the myocardial pattern of protein O-GlcNAcylation in each phase by immunoblotting using a pan-specific CTD 110.6 anti-O-GlcNAc antibody. The patterns varied among different phases. A number of bands in the immunoblotting showed a significant increase in the level of global protein O-GlcNAcylation was observed in VOed rats in both the CH and HF phases and VOed rats with VO correction in the HF phase (line 310–318).
Reviewer 2 Report
In the manuscript of Ying-Chang Tung et al. the authors seek to investigate the changes and reversibility of cardiac energy metabolism in VOed rat hearts with the combination of transcriptomic and metabolomic analyses. The technique used (aortocaval fistula surgery) is unknown to me, and trusting the literature, many others. I think the manuscript would greatly improve with a figure that shows the procedure graphically and the (hypertrophic) results on the heart by usingwhole heart stainings/fibrosis staining and combine this with the volume parameters of figure 1.
line 73: needed --> needle
Figure 2: Is there also overlap between CHcov and HFcov? Please rearrange the venn diagrams by showing one diagram with 4 leaves --> all inetractions/overlap.
Figure 3 is illegible in this format and should be altered. Furthemore, are the pathways highlighted in the manuscript: Glucose and Lipid metabolism, the most differentially expressed pathways? Is this something that could be shown by perfroming a pathway enrichement analysis using the complete data on diffreentially expressed genes (since the authors performed a genome wide expression analysis)? Additionally, the microarray data should be stored in a online repository (eg. GEO).
Were there differences observed in eating behavior between sham and VO animals? This could potentialy explain the changes in metabolites observed.
The authors discuss their surprising finding of reduced glucose uptake and glycolysis in the aortocaval fistula surgery-model. How does this compare to a more conventional TAC-model?
Author Response
Reviewer 2
In the manuscript of Ying-Chang Tung et al. the authors seek to investigate the changes and reversibility of cardiac energy metabolism in VOed rat hearts with the combination of transcriptomic and metabolomic analyses. The technique used (aortocaval fistula surgery) is unknown to me, and trusting the literature, many others. I think the manuscript would greatly improve with a figure that shows the procedure graphically and the (hypertrophic) results on the heart by using whole heart staining/fibrosis staining and combine this with the volume parameters of figure 1.
line 73: needed --> needle
Ans: We are sorry for the typo and have corrected it in the revised manuscript (line 74).
Figure 2: Is there also overlap between CHcov and HFcov? Please rearrange the venn diagrams by showing one diagram with 4 leaves --> all inetractions/overlap.
Ans: We have revised the Venn diagrams to illustrate the number of differentially expressed genes in all the four phases in one diagram (Figure 3C). The Venn diagrams showed that the number of differentially expressed genes involved in glucose and lipid metabolism increased as VOed hearts transitioned from CH to HF (line 207–209).
Figure 3 is illegible in this format and should be altered. Furthemore, are the pathways highlighted in the manuscript: Glucose and Lipid metabolism, the most differentially expressed pathways? Is this something that could be shown by performing a pathway enrichment analysis using the complete data on diffreentially expressed genes (since the authors performed a genome wide expression analysis)? Additionally, the microarray data should be stored in an online repository (eg. GEO).
Ans: We are thankful for the reviewer’s comment and have performed pathway enrichment analyses using the IPA software. The histograms of top 30 canonical pathways, ranked by -log (P value), in the CH, CHCOV, HF, and HFCOVphases are illustrated in Figures S1, S2, 3A, and 3B, respectively. To make the figure panel more readable, pathways maps of the HF and HFCOV phases are shown in Figure 4 in the manuscript and those of CH and CHCOV phases are shown in Figure S4. The microarray data has been submitted to GEO (accession number: GSE97044; https://www.ncbi.nlm.nih.gov/geo/query/acc.cgi?acc=GSE97044)(line 124).
Were there differences observed in eating behavior between sham and VO animals? This could potentially explain the changes in metabolites observed.
Ans: We did not specifically assess the amounts of chow diet the animals ate or any possible change in the feeding behavior in the animals in this study. How the changes in eating behavior may have influenced the results of metabolomic analysis is unknown. We have clarified this in the study limitations (line 455–457).
The authors discuss their surprising finding of reduced glucose uptake and glycolysis in the aortocaval fistula surgery-model. How does this compare to a more conventional TAC-model?
Ans: Animal models of transverse aortic constriction (TAC)-induced pressure overload have demonstrated that myocardial glucose uptake and utilization rates were increased and fatty acid oxidation was suppressed. As observed in the fetal hearts, increased glucose utilization in pressure overload was associated with increased insulin-independent GLUT1 and decreased insulin-dependent GLUT4 transporters. In contrast, our VO model exhibited no significant change in the expression of GLUT1 but the downregulation of GLUT4 and other genes that encode enzymes involved in glycolysis. Differences in experimental models and hemodynamic loads may account for the different metabolic adaptations between the pressure-overloaded and the VOed hearts (line 352–364).
Round 2
Reviewer 2 Report
I'm pleased with the improvement of the manuscript after the changes made by the authors
Author Response
There seems to be no new comment from Reviewer 2.
We are very grateful for the reviewers for helping us improve our manuscript.